# Investigation into the Effect of RFSSW Parameters on Tensile Shear Fracture Load of 7075-T6 Alclad Aluminium Alloy Joints

**DOI:** 10.3390/ma14123397

**Published:** 2021-06-19

**Authors:** Andrzej Kubit, Tomasz Trzepieciński, Elżbieta Gadalińska, Ján Slota, Wojciech Bochnowski

**Affiliations:** 1Department of Manufacturing and Production Engineering, Rzeszow University of Technology, al. Powst. Warszawy 8, 35-959 Rzeszów, Poland; 2Department of Materials Forming and Processing, Rzeszow University of Technology, al. Powst. Warszawy 8, 35-959 Rzeszów, Poland; tomtrz@prz.edu.pl; 3Materials & Structures Research Center, Łukasiewicz Research Network—Institute of Aviation, al. Krakowska 110/114, 02-256 Warsaw, Poland; Elzbieta.Gadalinska@ilot.lukasiewicz.gov.pl; 4Institute of Technology and Materials Engineering, Technical University of Košice, Mäsiarska 74, 040 01 Košice, Slovakia; jan.slota@tuke.sk; 5Department of Mathematics and Natural Sciences, University of Rzeszow, 1 Pigonia St., 35-310 Rzeszów, Poland; wobochno@univ.rzeszow.pl

**Keywords:** aluminium alloy, failure mechanism, fracture mode, RFSSW, welding

## Abstract

The aim of the investigations was to determine the effect of parameters of refill friction stir spot welding (RFSSW) on the fracture load and failure mechanisms of the resulting joint. RFSSW joints were made in 7075-T6 Alclad aluminium alloy sheets using different welding parameters. The load capacity of joints was determined under tensile/shear loadings. Finite element-based numerical simulations of the joint-loading process were carried out, taking into account the variability of elasto-plastic properties of weld material through the joint cross-section. The influence of welding parameters on selected phenomena occurring during the destruction of the joint is presented. The considerations were supported by a fractographic analysis based on SEM images of fractures. It was found that there is a certain optimal amount of heat generated, which is necessary to produce the correct joint in terms of its load capacity. This value should not be exceeded, because it leads to weakening of the base material and thus to a reduction in the strength of the joint. Samples subjected to uniaxial tensile shear load showed three types of failure mode (tensile fracture, shear fracture, plug type fracture) depending on the tool rotational speed and duration of welding. Prediction of the fracture mode using FE-based numerical modelling was consistent with the experimental results. The samples that were damaged due to the tensile fracture of the lower sheet revealed a load capacity (LC) of 5.76 KN. The average value of LC for the shear fracture failure mechanism was 5.24 kN. The average value of the LC for plug-type fracture mode was 5.02 kN. It was found that there is an optimal amount of heat generated, which is necessary to produce the correct joint in terms of its LC. Excessive overheating of the joint leads to a weakening of the base metal and thus a reduction in the strength of the joint. Measurements of residual stresses along the axis specimens showed the presence of stresses with a certain constant value for the welded area on the side of the 1.6 mm thick plate.

## 1. Introduction

The growing demand for light and easy-to-recycle alloys from the automotive and aerospace industries opens a window for a new use of aluminium alloys [1,2]. These alloys create difficulties when joined with conventional welding techniques because they create brittle intermetallic compounds in the microstructure that cause intermetallic cracks [3]. Solid state welding makes it possible to eliminate this drawback. One such technique, currently under development, is refill friction stir spot welding (RFSSW), developed and patented by Helmholtz-Zentrum Geesthacht, Germany [4]. Nevertheless, high strength aluminium alloys, i.e., AA7075, can be safely arc welded without hot cracks by introducing nanoparticle-enabled phase control during welding [5].

RFSSW is a variation of friction stir spot welding (FSSW) technique invented by The Welding Institute in 1991 [6]. RFSSW is an emerging technology that allows you to eliminate or limit riveting, which involves the need to make holes in the structure and add foreign material, thus increasing the weight of the structure. FSSW leaves behind a keyhole, which is a tool exit hole at the centre of the spot weld [7]. Keyhole is susceptible to corrosion and causes stress concentrations that degrade the structural properties of the weld under fatigue loading conditions. Unlike FSSW, RFSSW produces a spot joint with a near-flush surface finish that is free from a key or exit hole [7]. Furthermore, RFSSW requires low feed rates to enable sufficient plastification of the materials and optimum joint point formation [8].

For light weight joining, self-piercing riveting (SPR) is gaining popularity which also does not require pre hole [9]. In the SPR process, a rivet penetrates the top sheet and creates an interlock between the top and bottom sheets by flaring the legs without perforating the bottom sheet [10]. SPR has many advantages including low noise, no fume, no surface treatment required, no spark, ability to join multi-layer materials and mixed materials and ability to produce joints with high fatigue strength [11]. Compared with other joining techniques such as FSSW, RFSSW, resistance spot welding and clinching, SPR has proven to be an effective process to join dissimilar materials, from aluminium with composite to aluminium with advanced high strength steels [12]. Several of the key parameters that affect the quality of an SPR joint and some assistive technologies that have the potential of improving the quality of a joint were discussed by Haque [13].

Application of RFSSW for creating joints in similar AA6111-T4 automotive aluminium alloys and in dissimilar aluminium AA6111-T4 to magnesium AZ31-H24 alloy combinations has been documented by Al-Zubaidy [14]. The performance of the joints was investigated in terms of the effect of the welding parameters in allowing optimisation of the process for short welding-cycles. Schmal et al. [15] investigated the influence of RFSSW process parameters on joint formation and load capacity of the aluminium alloy joints. They developed an ultrasonic-based non-destructive testing method for evaluation concerning its suitability for RFSSW joints.

The literature on RFSSW mainly concerns the study of the influence of the process parameters on the resulting properties of the weld and its microstructure [16,17,18]. Okada et al. [19] examined RFSSW joints to evaluate their performance with regard to parameters such as tensile shear strength and strength of the heat-affected zone, with the use of various surface conditions. There are three process parameters that influence the load capacity and microstructure of the joint, i.e., spindle speed, welding time and tool plunge depth. Plunge depth had strong positive effects on the ultimate tension load, which means that more plunge depths produced stronger joints. Lower spindle speeds produce higher ultimate loads [20]. Rosendo et al. [21] showed that the geometrical features of AA6181-T4 aluminium alloy sheets played a very important role in the fracture mechanism and hence the mechanical performance of joints. Suhuddin et al. [22] indicated that tool rotational speed directly affects the process temperature and, as a consequence, the metallurgical reaction taking place at the joint interface. Kluz et al. [23] found that it was possible to choose the optimal welding parameters, taking into account the maximisation of load capacity and the minimisation of dispersion of joint strength. The analysis of the fatigue fracture of the RFSSW joint was carried out by Kubit et al. [24]. They revealed that the Alclad layer at the bottom of the weld is a kind of structural notch and, in this situation, can be the location of the initiation of fatigue cracking. The evaluation of crack initiation and growth at microscopic scale is a crucial issue for the safety assessment of macroscopical fractures. Maruschak et al. [25] demonstrated that increase in fatigue crack length is accompanied by the increase in the size of plastic deformation zone near the crack tip that makes the transition from the quasi-brittle failure to the fatigue one with the formation of fatigue striations. The elementary actions of plastic deformation resulting in crack are defined by processes related to the dislocation motion [26].

Despite the undisputed advantages of RFSSW technology in relation to joining difficult-to-weld metals, due to a number of factors determining the formation of a high-quality joint, the conducting of analyses and research aimed at gaining a better understanding of the mechanism of joint formation and destruction is still justified. Controlling the parameters of the welding process makes it possible to have many combinations of parameters which translate into different conditions and, consequently, a different quality of joint in terms of load-capacity and structural features. Although there are already many scientific publications dealing with the selection of appropriate RSFFW parameters, the guidelines for determining the optimal process parameters for thin-walled aluminium alloy structures have still not been clearly defined. The optimal values of welding parameters for a specific alloy may not be directly extended to other materials. Thermomechanical phenomena occurring during welding depend on the geometrical dimensions of the connected structures, the clad of metal sheets and the thickness of the joined elements. Due to the limited number of comprehensive works on the fracture load of Alclad sheets, the results of an analysis of the failure mechanism of an overlap joint of 7075-T6 Alclad aluminium alloy sheets subjected to the static tensile shear test are presented in this paper. The influence of welding parameters on selected phenomena occurring during the destruction of the joint are also discussed. The considerations were supported by finite element method (FEM) simulations, fractographic analysis based on SEM images of fractures and measurements of residual stresses occurring in the joint using X-ray diffraction (XRD).

## 2. Materials and Methods

### 2.1. Material

Alclad 7075-T6 aluminium alloy sheets with thicknesses of 1.6 mm and 0.8 mm were used as the test material for the fabrication of single-lap RFSSW joints. The joining configuration in single-lap joints of Alclad 7075-T6 sheets of different thicknesses corresponds to the joining configuration of skin plate (0.8-mm-thick) with a stiffening stringer (1.6-mm thick) in the typical aircraft structure. The basic mechanical parameters (Table 1) were determined by tensile testing according to the ISO 6892-1:2016 standard [27]. Three samples were tested and the average values of basic mechanical parameters were determined. The work hardening parameters (*K*—hardening coefficient and *n*—strain hardening exponent) were determined by the approximation of experimental stress-strain data using the power law equation, also known as the Hollomon equation: *σ* = *K*·ε^n^.

### 2.2. Refill Friction Stir Spot Welding

Welding of single-lap joints (Figure 1) was conducted using an RPS100 spot welder by Harms & Wende GmbH & Co KG (Hamburg, Germany). The RFSSW tool consists of three elements (Figure 2): a clamping ring, a sleeve with an external diameter of 9 mm and a pin with diameter of 5.2 mm. The RFSSW welding process can be divided into four stages (Figure 2): touchdown, plunging, refilling and tool retraction. The stages of the RFSSW process are as follows [24]:Both the sleeve and the pin start to rotate and rub on the sheet for a while to heat the material;The sleeve and pin move downward and upward, respectively, and therefore the plasticised material displaced by the sleeve is squeezed into the cylindrical cavity created by the upward movement of the pin;Both the sleeve’s and the pin’s directions of movement begin to reverse, and the plasticised material in the cylindrical cavity is squeezed back by the sleeve;The welding head is retracted from the working area.

Based on recent papers by the authors [23,24], RFSSW joints were fabricated at a constant tool plunge depth of 1.5 mm. Various values of the welding time were adopted in the research plan. This was assumed because welding time is directly related to the amount of heat generated in the joint. The temperature resulting from the welding process, which interacts over a longer period of time, allows the material to be recrystallised as well as the removal of any structural notches caused by the penetration of the tool sleeve. If the welding time is too long, this leads to unfavourable phenomena in the HAZ manifested by a sharp reduction in the load capacity of joints. The choice of the RFSSW process parameters therefore requires compromise solutions, i.e., the selection of tool rotational speed at a level ensuring adequate plasticisation of the base material and to obtain the required load capacity of the joint and then a welding duration ensuring that the appropriate joint microstructure is obtained, thus conditioning the required process repeatability. The plan of the experiments is listed in Table 2.

Penetration of the upper sheet by the pin to a depth of 1.5 mm ensures optimal mixing of the materials of both sheets and the highest load capacity [23,24]. The decrease in the strength of the joint with the increase in tool penetration up to the value of 1.5 mm is caused by the weakening of the lower part of the joint by the operation of the sleeve [23,24].

### 2.3. Tensile Shear Test

The static strength tests were carried out at room temperature on a Zwick/Roell Z100 (ZwickRoell GmbH & Co.KG, Ulm, Germany) testing machine. The tensile speed was 5 mm/min. The shear tests were conducted according to the EN ISO 14273 [28] standard. The samples (Figure 1) were attached to the grippers of the testing machine for a length of 30 mm.

### 2.4. Microstructural Analysis

The morphologies of the fracture surfaces of the RFSSW joints were examined using an S-3400 Phenom ProX scanning electron microscope (Nanoscience Instruments, Phoenix, AZ, USA).

### 2.5. Microhardness Testing

Zones with different grain sizes can be observed in the cross-section of the RFSSW. It is already known [29,30,31] that these produce different mechanical properties in these zones. These regions are the designated stir zone (SZ), heat affected zone (HAZ), thermo-mechanically affected zone (TMAZ) and base metal (BM). A detailed description of their characteristics can be found in previous papers by the authors [23,27]. It is possible to easily identify the approximate boundary between the BM, HAZ, SZ and TMAZ in the macrographs, as shown in Figure 3. A bright contrasting bonding ligament is found embedded in the boundary between the materials of the upper and lower sheets. The bonding ligament in the case of 7075-T6 aluminium alloy sheets is composed of 7072 aluminium alloy (99 wt % Al, 1 wt % Zn). The change in grain size due to thermo-mechanical phenomena in the nugget during welding has effects on the variation of the mechanical properties of the material through the weld. The purpose of measuring microhardness was to determine the mechanical properties of the weld material in order to define the numerical model of the RFSSW joint. Microhardness tests were performed for a weld made with the parameters: tool rotational speed n = 2400 rpm and duration of welding t = 3.5 s (experiment E2). Microhardness was used to quantify changes in the mechanical properties of the material in individual zones. To determine the hardness of the material and Young’s modulus in specific zones of the weld (Figure 3), a Micro-Combi (CSM Instruments, Peseux, Switzerland) tester with a Vickers pyramidal diamond tip was used with a load of 1000 mN. In order to assure the statistical reproducibility of the measurements, six indentations were made for each zone. Microhardness HV0.1 was determined according to the Oliver–Pharr method. The loading rate and the unloading rate were both equal to 33.33 mN/s.

### 2.6. Numerical Modelling

Finite element-based numerical modelling of the panel deformation process was carried out in the ABAQUS program. The geometric model (Figure 4) of the single-lap RFSSW joint corresponded to the real dimensions of the test object (Figure 1). Due to the symmetry of the single-lap joint, the program considered half of the model with the appropriate symmetry conditions. The model omits the consideration of thin clad layers. These thin layers would require a dense mesh, which would significantly increase the computational time. In addition, the clad layers on sheet surfaces, due to their much lower strength and thickness in relation to the BM, carry practically no load. Therefore, the thin clad on the sheet surfaces does not have a significant effect on the strength of the joint. The bonding ligament (Alclad) in the weld nugget was of course included in the analysis.

The weld area was divided into areas corresponding to the division of critical zones of the RFSSW weld (Figure 4): BM, SZ, HAZ, TMAZ and the bonding ligament, based on the experimental results shown in Figure 3. Appropriate boundary conditions were assigned in the areas covered by the grippers of the testing machine. The displacement of the end of the 1.6 mm thick sheet was blocked at a length of 30 mm in all x, y and z directions (U1 = U2 = U3 = 0 in Figure 5). The displacement of the end of the 0.8 mm thick sheet was also fixed in the x and z directions. A displacement in the y direction (U1 > 0) was simultaneously applied to this area.

During analysis, 4-node C3D4 elements were used for model discretisation. A denser mesh was used in the vicinity of the weld (Figure 6). Based on a mesh sensitivity analysis for different element sizes in the weld region, the analysis was conducted with about 0.9 million elements in the FE-based model.

An elastic-plastic material model with a strain-hardening phenomenon was adopted according to the results in Table 2. The elastic properties of the material in the zones of the 7075-T6 aluminium alloy weld determined using the Oliver–Pharr method are listed in Table 3. The mechanical properties of Alclad were assumed based on the literature [32]: Young’s modulus *E* = 68,000 GPa, Poisson’s ratio *ν* = 0.33, yield stress *σ_y_* = 68 MPa. The value of Poisson’s ratio for 7075-T6 aluminium alloy was set at *ν* = 0.33 [33]. The sheet material was defined as isotropic according to the von Mises criterion. The yield stress in individual zones of weld was determined on the basis of the microhardness value. By considering the strain-hardening exponent, Cahoon et al. [34] formulated an empirical equation for elastic-plastic materials. Based on the linear relation between hardness and field strength [34], the equation for the determination of yield stress σ_y_ is given by
(1)$$\sigma_{y}=HV\times \frac{{\left(0.1\right)}^{n}}{3}$$

The load-displacement curves of Vickers microhardness indentation of the material in different zones of the RFSSW joint are shown in Figure 7. Hysteresis loops are typical of materials with elastic-plastic characteristics. The material zones exposed to the RFSSW tool reaction show increased strength compared with the BM. Consequently, the indentation depths corresponding to the maximum loading force registered for HAZ, TMAZ and SZ are smaller. The processes taking place during the thermo-mechanical formation of the joint also affect the value of Young’s modulus determined with the use of the Oliver–Pharr method (Table 3) because they change the angle of inclination of the unloading curve.

### 2.7. X-ray Diffraction Measurements

The aim of the X-ray diffraction measurements was to experimentally determine the residual stress values around the joints after the RFSSW process performed with a different set of parameters. The following variants were selected for this analysis: E5 and E6. E5 specimen was welded with spindle speed equal to 2200 rpm, with tool plunge depth—1.5 mm and with the welding time—2.5 s. In the case of E5 specimen, the spindle speed was 2800 rpm, the tool plunge depth was the same—1.5mm and the time of welding was shorter and equal to 1.5 s. With this choice of welding parameters for specimens dedicated to residual stress measurements, it will be possible to observe the effect of increasing the pin and sleeve rotation frequency during welding at the expense of the duration of the entire process.

The X-ray diffraction measurements were carried out for 20 subsequent points along the sample axis starting from the centre of the weld. The measuring spot for each point had a diameter of 1 mm and the distance between measuring points was also 1 mm.

All diffraction measurements were performed with a portable Xstress 3000 G2R diffractometer which is a device dedicated for stress measurements with standardised sin^2^ψ method [35,36]. This specific method is described in detail in the framework of EN 15305:2008 [37] standard, and all stress measurements were therefore carried out under the guidance of this standard. Due to the fact that X-rays are strongly absorbed by metallic materials, the measurements only concern the surface layer of the specimens. On the other hand, the fact that the phenomenon of X-ray diffraction separates the information coming from different phases of the investigated alloy, the experiments carried out in this part of the work were performed for the aluminium phase. In the experiment, K_α_ radiation of the chromium X-ray tube was used and the measurements were carried out for the tangential and radial directions in relation to the geometry of the welds; these directions were illustrated in Figure 8. All measurements were performed for both sheets of the specimen E5 and for thicker sheet of the E6 specimen. 

A reflection with Miller indices of (311) was used to determine the stress level by X-ray diffraction. The value of Braggs’ angle 2θ for this reflection is approximately 139.3°. The duration of collecting statistical information for one reflection was 300 s, and measurements for 13 diffraction reflections in the range of ψ angle values from −40° to 40° were used for one stress measurement. In order to obtain the most accurate distribution around the 2θ angle, the oscillation option around the mean value of this angle was used; the oscillation value was 5°. The residual stresses were measured in the subsurface layer at a depth of about 50 micrometers.

The values of Young’s modulus E = 70.6 GPa, Poisson’s ratio ν = 0.345, averaged for the aluminium phase, taken form the Xstress300 G2R software libraries [38], were used to calculate the values of stresses measured by X-ray diffraction. The cross-correlation method was used to determine the position of diffraction peaks.

## 3. Results and Discussion

### 3.1. Tensile Properties

Figure 9 shows representative tensile curves for the variants of joints considered. It was shown that, depending on the parameters of the welding process, the destruction proceeds according to various mechanisms. Basically, three modes of joint destruction are distinguished here, i.e., cracking of the lower sheet as a result of its stretching after its previous deflection (Figure 10a), shear of the weld nugget in the plane between the lower and upper sheets (Figure 10b) and destruction of the weld by plug-type fracture (Figure 10c). These mechanisms will be discussed in the next chapters. The weld produced at the tool rotational speed of n = 2400 rpm and welding time t = 3.5 s (experiment E2) had the highest load capacity. At the same time, the greatest elongation connected with the sample crack in the BM was observed in the specimen (Figure 10a). Figure 11 shows ultimate tensile shear strength corresponding to specific fracture modes.

### 3.2. Analysis of the Tensile Fracture Mechanism

The samples that were damaged due to the tensile fracture of the lower sheet were prepared using the following parameters: tool rotational speed n = 2400 rpm and welding time t = 3.5 s (experiment E2). The highest load capacity, with an average of 5.78 kN, and the smallest dispersion (SD: 47.20 N) were found in this group of joints. The mode of joint destruction demonstrated proves the high strength of the weld nugget which was not sheared. The crack in the tensioned lower sheet (Figure 12) is not the result of the stress reaching the *R_m_* value of the BM.

Directly in the vicinity of the weld edge, the sheet was locally weakened as a result of thermo-mechanical phenomena, exhibiting lower strength parameters compared with the BM material in the T6 state. Figure 12a,b shows, on a macroscopic scale, a sheet fracture forming directly at the periphery of a weld, while Figure 12c,d shows views of the edge of the fractured sheet at two different locations. It can be observed that, in the initial stage, the cracking took place by partial plasticisation of the material, while in the later stage, brittle cracking was observed.

In the cross-section of the fracture surface, it can be observed that the crack runs in a plane oblique to the surface of the lower sheet inclined at an angle of about 45°, as shown in Figure 13a–c, which shows SEM micrographs of the fracture edge in the vicinity of the opposite surfaces of the lower sheet. Areas of plastic and brittle fracture can be observed in the fracture surface.

It has been shown that the tensile fracture mode also occurs for joints fabricated with the use of welding parameters: tool rotational speed n = 2800 rpm and welding time t = 3.5 s (experiment E1). However, in this case, a significantly lower load capacity of the joints of 4.83 kN (SD:397.23 N) was observed. Metallographic analyses demonstrated the cause of the weakening of the joint. The reason for the weakening of the joint is the excessive area of the HAZ, which is caused by incorrectly selected friction conditions. The highest of the values of tool rotation speed and welding time considered lead to the generation of an excessive amount of heat in the joining process, which in turn leads to a significant extension of the HAZ (Figure 14). An additional phenomenon is the heat dissipation through the Alclad, because it has a much higher coefficient of thermal conductivity (229 W/mK) than the 7075 aluminium alloy (134 W/mK). This leads to the concentration of the expanded HAZ region in the central region of both lower and upper sheets. This in turn leads to a weakening of the BM, resulting in the cracking of the sheet metal at lower load values.

Therefore, it can be concluded that, in the RFSSW welding process of 7075-T6 Alclad aluminum sheets with the thicknesses and welding configurations presented, there is an optimal amount of heat generated in the welding process. Achieving this amount of heat is necessary to produce the correct joint in terms of its load capacity. On the other hand, too much heating of the material leads to a weakening of the base material and thus a decrease in the strength of the joint. This phenomenon is presented as an illustrative diagram in Figure 15. As the heat generated during welding is closely correlated with the rotational speed of the tool and the welding time, these parameters should be correctly selected for joining a given structure to ensure the highest load capacity of the joint.

The failure mechanism of the joint analysed is strictly dependent on the configuration of the sheets to be joined. As the sheets have two different thicknesses, a significant difference in their stiffness leads to an asymmetric state of stresses when loading the lap with an axial force *F*. Figure 16 shows the stress states that arise in the joint during its loading with a uniaxial force.

In terms of elastic strains, it can be assumed that the upper sheet, due to the twofold difference in thickness compared with the lower sheet, undergoes only negligibly small deformations; therefore, considerations were made regarding the stress distribution in the area of the lower sheet characterised by significantly lower stiffness. Bending stresses $\sigma_{y2}$ appear in the area of the weld in the lower sheet, as shown graphically in Figure 17.

The value of the stress below the yield stress of the material can be determined using Equation (2) resulting from the static equilibrium of the bending moments with regard to the z-axis.
(2)$$\sigma_{y2}=\frac{F_{yield,\sigma_{y2}}\times \left(t_{1}+t_{2}\right)\times I_{2}}{d}\leq f_{yield}$$
where *t*_1_ and *t*_2_ are thicknesses of the upper and lower sheet, respectively, *d* is the diameter of the weld nugget and *I*_2_ is the moment of inertia of the weld cross section.
(3)$$I_{2}=\frac{\pi \times d_{2}^{4}}{64}$$

In addition to bending stresses, compressive and tensile stresses in the direction of the x-axis act around the weld nugget both in the upper and lower sheets (Figure 18). The maximum value of these stresses can be determined according to Equation (4), while any value of stress in polar coordinates can be determined by Equation (5).
(4)$$\sigma_{x2,\;\;max}=\frac{4\times F}{n\times t_{2}\times d}\leq f_{yield}$$
(5)$$\sigma_{x2}\left(\theta \right)=\sigma_{x2,\;\;max}\times cos\theta \leq f_{yield}$$

On the basis of the equilibrium of stresses in the weld nugget, it is possible to determine the value of the maximum tensile force *F* of the lap specimen causing the elastic deformation of the lower sheet. Internal stresses acting on the weld nugget are presented graphically in Figure 17. According to the condition of the equilibrium of stresses for the load distribution analysed, the following relations can be written:(6)$$2\times \sigma_{x2}\times t_{2}\times d=2\times \sigma_{x1}\times t_{1}\times d$$
(7)$$\sigma_{x2}=\sigma_{x1}\times \frac{t_{1}}{t_{2}}$$
(8)$$\sigma_{x2}=\frac{F}{2\times t_{1}\times d}\times \frac{t_{1}}{t_{2}}$$
(9)$$\sigma_{x2}=\frac{F}{2\times t_{2}\times d}\geq \sigma_{yield}$$

On the basis of the equilibrium condition of the bending moments, the value of the stress $\sigma_{y2}$ can be determined from the following formulae:(10)$$2\times \sigma_{x1}\times t_{1}\times d\times \left(\frac{t_{1}}{2}+\frac{t_{2}}{2}\right)=\sigma_{y2}\times 2\times \frac{I_{2}}{d}$$
(11)$$\sigma_{x1}\times t_{1}\times d\times \left(t_{1}+t_{2}\right)=\sigma_{y2}\times \frac{\frac{\pi }{32}\times d^{4}}{d}$$

Taking into account, in Equation (11), that
(12)$$\sigma_{x1}=\frac{F}{2\times t_{1}\times d}$$
we can write
(13)$$\frac{F}{2\times t_{1}\times d}\times t_{1}\times d\times \left(t_{1}+t_{2}\right)=\sigma_{y2}\times \frac{\pi }{32}\times d^{3}$$
and
(14)$$\sigma_{y2}=\frac{16\times F\times \left(t_{1}+t_{2}\right)}{\pi \times d^{3}}\geq \sigma_{yield}$$

On the basis of the values of the stresses $\sigma_{x2}=\frac{F}{2\times t_{2}\times d}$ and $\sigma_{y2}$ (Equation (14)), it is possible to determine the reduced stress for the lower sheet:(15)$$\sigma_{Eq2}=\sqrt{{\sigma_{x2}}^{2}+{\sigma_{y2}}^{2}}=\sqrt{{\left(\frac{F}{2\times t_{2}\times d}\right)}^{2}+{\left(\frac{16\times F\times \left(t_{1}+t_{2}\right)}{\pi \times d^{3}}\right)}^{2}}$$
(16)$$\sigma_{Eq2}=F\times \sqrt{{\left(\frac{1}{2\times t_{2}\times d}\right)}^{2}+{\left(\frac{16\times \left(t_{1}+t_{2}\right)}{\pi \times d^{3}}\right)}^{2}}\geq \sigma_{yield}$$

The destructive force of the lower sheet can therefore be determined on the basis of the following relationship:(17)$$F_{ultimate2}=\frac{\sigma_{ultimate}}{\sqrt{{\left(\frac{1}{2\times t_{2}\times d}\right)}^{2}+{\left(\frac{16\times (t_{1}+t_{2})}{\pi \times d^{3}}\right)}^{2}}}$$

### 3.3. Analysis of the Shear Fracture Failure Mechanism

Insufficient plasticisation of the joined materials during the welding process results in the fabrication of a relatively weak joint. Joints that were damaged by shear fracture were fabricated using the following parameters: tool rotational speed n = 2000 rpm, welding time t = 2.5 s (experiment E3). The average value of the load capacity for this fracture mode was 5.24 kN (SD: 87.31 N). A similar form of joint fracture was demonstrated for the welded variant using the following parameters: tool rotational speed n = 2200 rpm, welding time t = 2.5 s (experiment E5). In this case, however, a complex form of failure occurred, a crack occurred along the perimeter of the weld in the lower sheet. For this fracture mode, average load capacity was 5.67 kN (SD: 69.27 N). Shear fracture failure was also demonstrated for the joint fabricated with the use of the following parameters: tool rotational speed n = 2200 rpm, welding time t = 2.5 s (experiment E6). But at the moment of failure in this variant, the lower sheet was significantly deflected, which resulted in significant components of normal stresses causing peel. Relatively low load capacity and large dispersion of results were obtained for this mode, and the average value of the load capacity was 4.81 kN (SD: 147.15 N).

In order to be able to more precisely define the phenomena occurring at the joint shear, the fracture surfaces were analysed using SEM (for experiment E3). On the fracture surface of the specimen, which was damaged in shear fracture mode, several basic areas can be distinguished (Figure 19a). These areas can be associated with characteristic zones created during joint formation. In the final stage of welding, a pin in the central part of the weld presses the plasticised material, exerting high pressure. Under the influence of the high pressure and temperature in the area below the periphery of the pin, the phenomenon of dynamic recrystallisation becomes most intense. Hence, in this area, the microstructure is fine-grained with the smallest grain sizes. This translates into the formation of the strongest joint in the area of influence of the pin. In addition, in this area, the clad material in the weld nugget is lifted above the contact surface of the upper and lower sheet. Therefore, the central area of the weld was subjected to destruction of the fine-grained structure of the 7075-T6 aluminium alloy. Based on the view of the fracture, it can be concluded that plastic cracking occurred here. 

In the sleeve interaction zone, the area furthest from the joint axis was formed by the material flowing onto the periphery of the weld at the end of the process. In this case, the clad material exists in the joint plane, and the shear plane runs along the clad boundary and the 7075-T6 alloy. Typically, ductile cracking is observed, characterised by the formation of pits and craters (Figure 19b–e) as a result of material flow; the cracking phenomenon occurred here by nucleation and growth of voids.

The area most distant from the weld axis (Figure 19f,g) is characterised by a relatively weak bond of the material pressed by the pin in the final phase of the RFSSW process with the substrate. This is a result of the fact that the weld circumference zone during the welding process is subjected to a lower effect of heat and pressure compared with the sleeve interaction zone. Joint fracture is initiated in the area where the joint is weakest.

### 3.4. Analysis of the Plug-Type Fracture Mode

Destruction of the joint through plug-type fracture may result from a defect in the weld in the form of a weakened perimeter of the weld. This mode of fracture was obtained for samples produced using the following welding parameters: tool rotational speed n = 1800 rpm and welding time t = 1.5 s (experiment E4). The average value of the load capacity for this variant was 5.02 kN (SD: 129.07 N).

The low value of the welding time as well as the tool’s rotational speed led to a situation in which friction did not generate enough heat to plasticise the material of the upper sheet. Hence, the operation of the sleeve led to the punching of the material, while in the final stage of welding, the insufficiently plasticised material did not get mixed with the cut edge. As a result, the weld structure is characterised by insufficient material mixing along its perimeter in the structure of the upper sheet, as shown in Figure 20, in which such phenomena as incomplete refill and a kissing bond can be observed. 

The joint defect indicated, after exceeding the specified value of the bending moment occurring during the tensile shear test, leads to tearing of the weld nugget out of the upper sheet. As shown in other works by the authors [24], the defect discussed has a particularly negative effect on the fatigue failure mechanism. It has been shown that joints with such a defect are destroyed in the high-cycle fatigue test due to stretching of the upper (thicker) sheet with a much smaller number of fatigue cycles than is the case with a correctly fabricated weld. Therefore, it is the cause of a significant decrease in fatigue life.

### 3.5. Numerical Modelling

Figure 21 shows the distribution of equivalent von Mises stresses on the longitudinal section of the joint at various stages of the sample stretching process.

The resistance of the RFSSW weld to the displacement of the sheets under tensile shear loading causes twisting due to the resulting bending moment (Figure 21a). In the next stages of the stretching process, this effect becomes more marked. The greatest stresses are concentrated near the edge of the weld. As the tensile load increases, these areas expand to the full thickness of the lower sheet. At the same time, the upper sheet is only highly stressed on the side of the fixed edge. On the opposite side of upper sheet, an area with compressive stress is visible as a result of the bending moment. In the area of the joint, the Alclad layer is clearly visible, which, due to its low strength, does not transfer significant loads.

The value of stresses on the surface of the upper sheet (Figure 22) is mainly related to the bending moment effect and, as proved in Figure 21, this only occurs at a certain depth. 

The crescent-shaped zone tends to a circumferential widening symmetrical to the direction of the joint load. The observations obtained in the investigations are consistent with the results of other researchers’ analyses of the tensile properties of a single-lap RFSSW joint of various materials [39,40].

After propagating the area of highest stress concentration through the thickness in the lower sheet (Figure 21) near the centreline of the weld, this area tended to propagate around the joint. In the lower sheet, on the side of the weld bottom, there is a clear ring-shaped character of the distribution of equivalent stresses (Figure 23). As the load on the sample increases, this area widens into the lower sheet area on the side of the loaded end of the sample. At the same time, the unloaded edge of the lower sheet becomes semicircular due to the bending of the free end of the lower sheet. The annular concentration of stresses around the weld in the lower sheet resulting from the difference in thicknesses of both lower and upper sheets influences the development of failure of the RFSSW joint.

The distribution of equivalent plastic strain magnitude (Figure 24b) provides evidence for the path of the joint destruction that was observed experimentally (Figure 24a). The distribution of the equivalent plastic strain magnitude (Figure 24b) shows a character directly related to the zone of the highest reduced stresses in the lower sheet (Figure 23). The area that was plastically deformed shows the same behaviour of propagation of the deformation gradient as in the stress distribution (Figure 23). Cracks are initiated in the lower part of the lower sheet located on the side of action of the tensile force (Figure 25). On the opposite side of the weld axis, a plastic deformation of Alclad occurs; however, the value of these deformations is not decisive in the location of the material destruction site. The Alclad is surrounded by the SZ and TMAZ zones, the material of which is much stronger than that of Alclad. This conclusion is in line with the results of Al-Zubaidy [14], who experimentally confirmed that weld defects in the centre of the joint do not significantly affect the joint strength.

### 3.6. Analysis of Residual Stress Values with X-ray Diffraction

The results of the stress measurements are presented in Figure 26, Figure 27 and Figure 28. Figure 26 shows the results of stress measurements for specimen E5, for a sheet 1.6 mm thick; Figure 27 shows the results of stress measurements for specimen E5, for a sheet 0.8 mm thick; while Figure 28 shows the results for specimen E6 from the side of the thicker plate, 1.6 mm thick. In the case of each of the figures, the curve (a) refers to the stress values for the tangential direction relative to the weld geometry, while the plot (b) provides information on the distribution of stresses relative to the weld centre for the radial direction of stresses. Additionally, in Figure 29, the results of measurements of the so called full width at half maximum (FWHM) of diffraction peaks for each of the measured sheets of each sample are presented (Figure 29a for 1.6 mm sheet, specimen E5; Figure 29b for 0.8 mm sheet, for specimen E5; Figure 29c for 1.6 mm sheet, for specimen E6). The half-width of the diffraction peaks can indirectly provide the information on the occurrence of microstrains between grains of the material or changes in the degree of its defectiveness.

For each specimen, for each plate and for each stress direction tested, the stress values are similar and range from about −60 MPa to about 40 MPa. In the area of the weld, much smaller errors in the measurement of stresses are visible, which are related to the better, from the point of view of the diffraction experiment, measurement statistics: the increased fineness of the material grains within the weld, observed as a result of other experiments carried out within the scope of this work, makes it possible to obtain diffraction information from a larger number of grains. Comparison of the stress results obtained for specimen E6 from the side of the sheet with a thickness of 1.6 mm (Figure 26a) and from the side of the sheet with a thickness of 0.8 mm (Figure 27a), for the tangential direction of stresses, leads to the conclusion that the values of stresses in the central part of the weld for the thinner plate are higher and are of a compressive nature (in the central part of the weld, they take the value of approx. −45 MPa). For the thinner sheet, the compressive stress values decrease and change to tensile at a distance of about 3 mm from the centre. Due to the significant relative error in the results obtained for the native material of the plate, it must be stated that the stress values on both sides of the specimen are similar and oscillate around 0 MPa. In the case of stresses in the radial direction, for both the thicker (Figure 26b) and the thinner (Figure 27b) sheets, the stress values in the central area of the weld are similar, being compressive and approx. −40 MPa. Moreover, in the case of this stress component, for the thinner sheet, there is a change in the stress value at a distance of approx. 3 mm from the centre of the weld. This change does not occur with the thicker plate. 

Comparison of the results for specimens E5 and E6, in which the formation of the RFSSW joint was carried out using different welding parameters, makes it possible to observe that increasing the number of pin and sleeve rotations at the expense of welding process duration does not significantly change the value of the tangential stresses (Figure 26a vs. Figure 28a) in the welded area. In the case of radial stresses, such a change in parameters may lead to the relaxation of compressive stresses that occurred in the case of specimen E5 (Figure 26a vs. Figure 28a).

Very similar results were found for the SPR joints [41]. The stresses found in the rivets were compressive in nature with a lower magnitude in the rivet head than in the rivet leg. A compressive behaviour of the residual stress was found at the centre in the sheet material inside the rivet bore. However, the magnitude of the compressive residual stresses started to decrease away from the centre and eventually the stresses became tensile at a distance of 3–5 mm from the rivet centre.

Comparison of the half-width values obtained by peak diffraction measurement (Figure 29) does not indicate significant differences in the distribution of this value for any of the sheets or specimens, and, therefore, there is no reason to suspect that changes in micro-strain or changes in dislocation density have occurred as a result of changes in the welding parameters. The significant change in the FWHM value for the point 5 mm away from the weld centre of specimen E5, for a 1.6 mm sheet (Figure 29a), is probably related to the uneven surface of the specimen in this area.

## 4. Conclusions

In this paper, the effect of RFSSW parameters on tensile shear fracture load of 7075-T6 Alclad aluminium alloy joint is investigated. Based on the results of the investigations, the following conclusions can be drawn:The sample produced at the tool rotational speed n = 2400 rpm and duration of welding t = 3.5 s had the highest load capacity. The results of the tensile shear test revealed tensile fracture mode for this specimen.As a result of insufficient plasticisation of the materials beeing joined during the RFSSW process with the parameters n = 2000 rpm and t = 2.5 s, a joint was produced, which showed the shear fracture mode. Microstructural analysis showed a fine-grained structure in the area of operation of the pin; however, the peripheral zone of the weld was subjected to less heat, which was associated with a weakening of the joint. The fracture initiated in this zone.Destruction of the weld through plug-type fracture mode is a weld defect associated with a weakened or insufficient mixing of material near the circumference of the weld in the upper sheet.The samples that were damaged due to the tensile fracture of the lower sheet were showed load capacity of 5.76 KN. The average value of load capacity for the shear fracture failure mechanism was 5.24 kN. The average value of the load capacity for plug-type fracture mode was 5.02 kN.The highest values of (i) tool rotation speed as well as (ii) welding time that were considered lead to the generation of an excessive amount of heat in the joining process, which in turn led to a significant expansion of the HAZ in both upper and lower sheets.It was found that there is an optimal amount of heat generated, which is necessary to produce the correct joint in terms of its load capacity. Excessive overheating of the joint leads to a weakening of the BM and thus a reduction in the strength of the joint.The processes taking place during thermo-mechanical joint formation change the value of the modulus of elasticity and the hardness of the weld nugget in relation to the BM.Measurements of residual stresses along the axis specimens showed the presence of stresses with a certain constant value for the welded area on the side of the 1.6 mm thick plate. In the case of thinner sheet (0.8 mm), the stress value relaxed at a distance of about 3 mm from the centre of the weld. Precise determination of the stress values in the plate’s base material was disabled by the relatively coarse grain size of the material.For each specimen, for each plate and for each stress direction tested, the stress values are similar and range from about −60 MPa to about 40 MPa.In the case of stresses in the radial direction, for both the thicker and the thinner sheets, the stress values in the central area of the weld are similar, being compressive and approx. −40 MPa.It is worth considering more in-depth diffraction studies to determine the full stress tensor and to determine von Mises stress values to make the comparison with numerical modelling possible.

## Figures and Tables

**Figure 1 materials-14-03397-f001:**
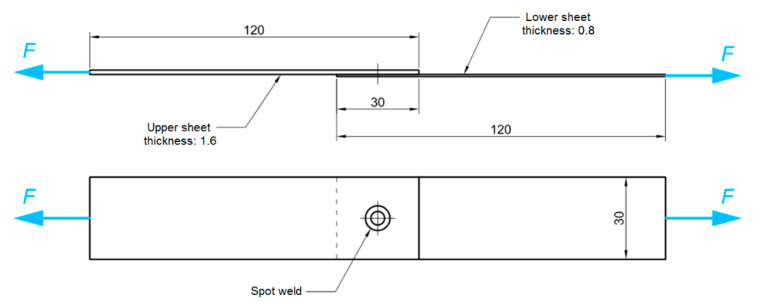
Dimensions (in mm) of the specimen for tensile/shear testing.

**Figure 2 materials-14-03397-f002:**
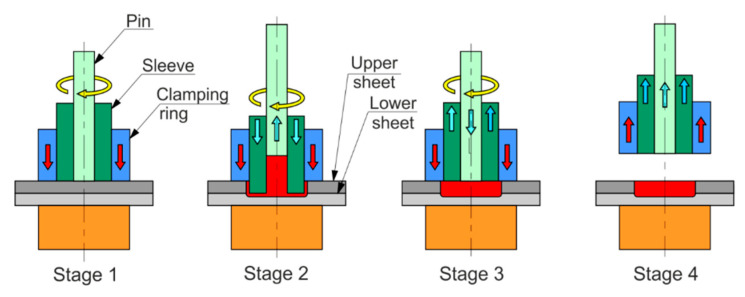
Stages of refill friction stir spot welding.

**Figure 3 materials-14-03397-f003:**
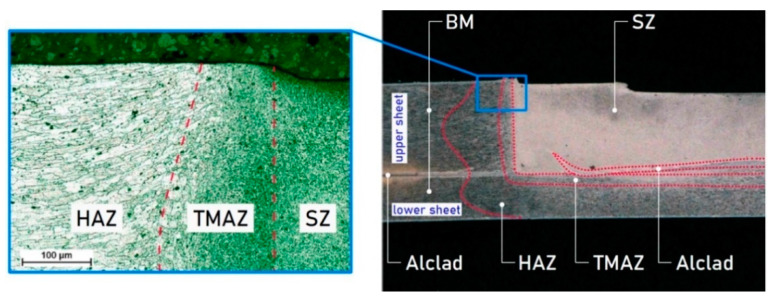
Critical zones of the FSSW joint (experiment E2).

**Figure 4 materials-14-03397-f004:**
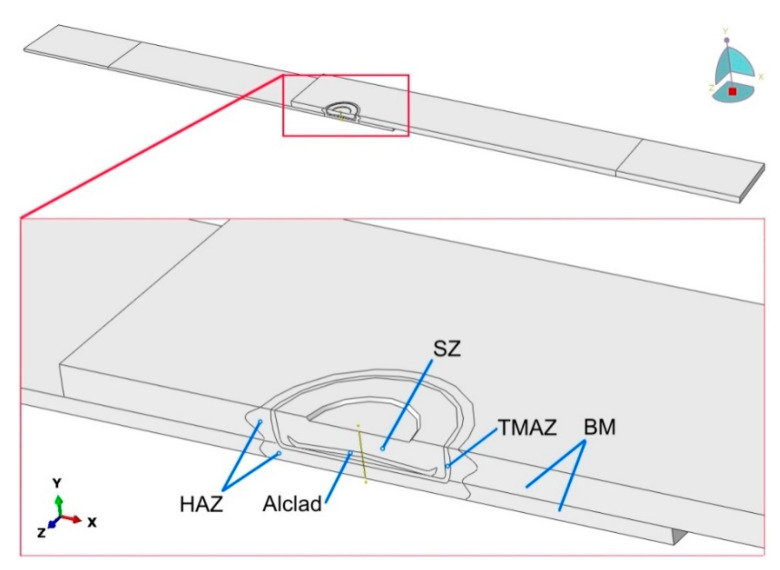
Model of the single-lap RFSSW joint.

**Figure 5 materials-14-03397-f005:**
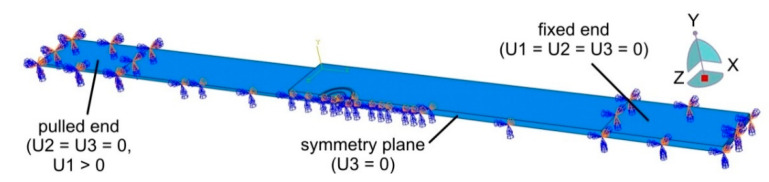
Boundary conditions in the model of the single-lap RFSSW joint.

**Figure 6 materials-14-03397-f006:**
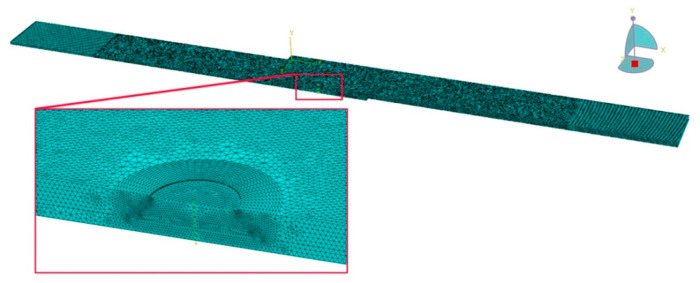
Finite element mesh of a single-lap RFSSW joint.

**Figure 7 materials-14-03397-f007:**
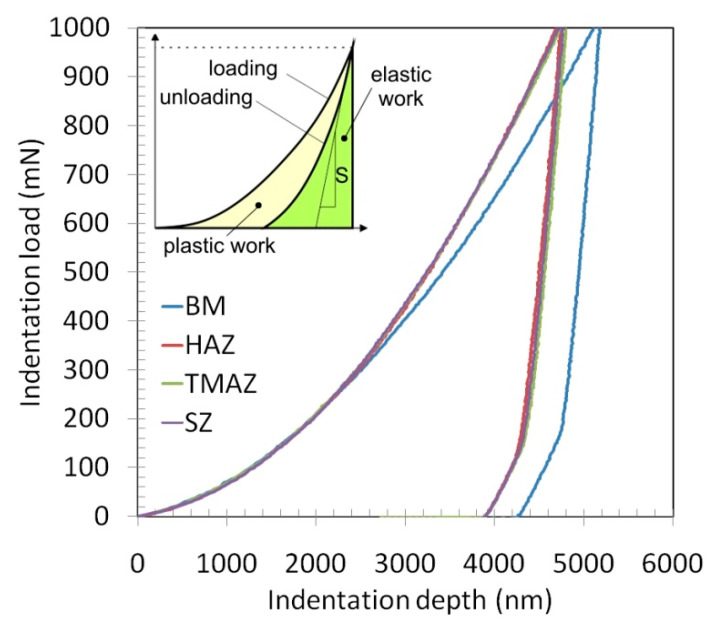
Indentation curves for different zones of RFSSW joint (experiment E2).

**Figure 8 materials-14-03397-f008:**
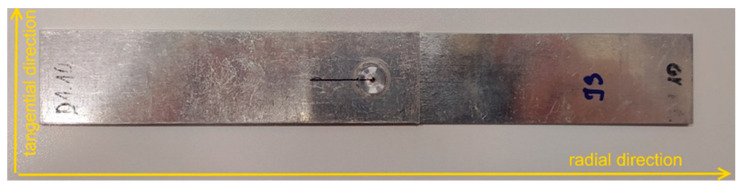
View of a sample of a lap joint with the directions of stress measurement marked.

**Figure 9 materials-14-03397-f009:**
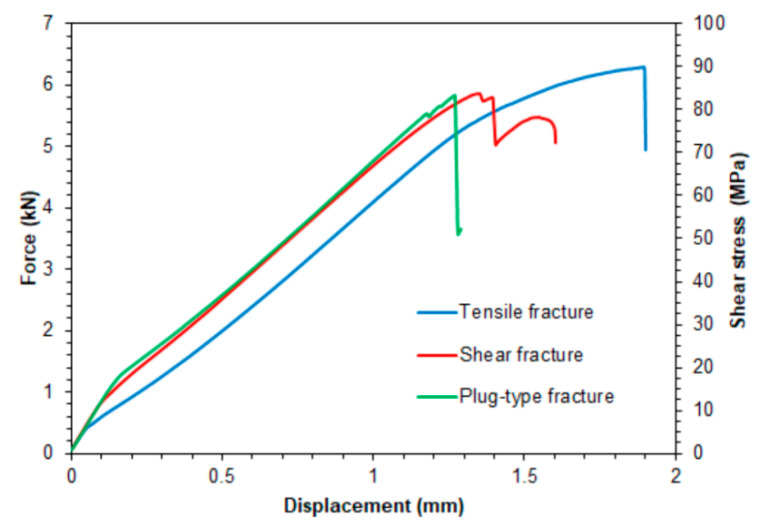
Force-displacement curves corresponding to specific fracture modes.

**Figure 10 materials-14-03397-f010:**
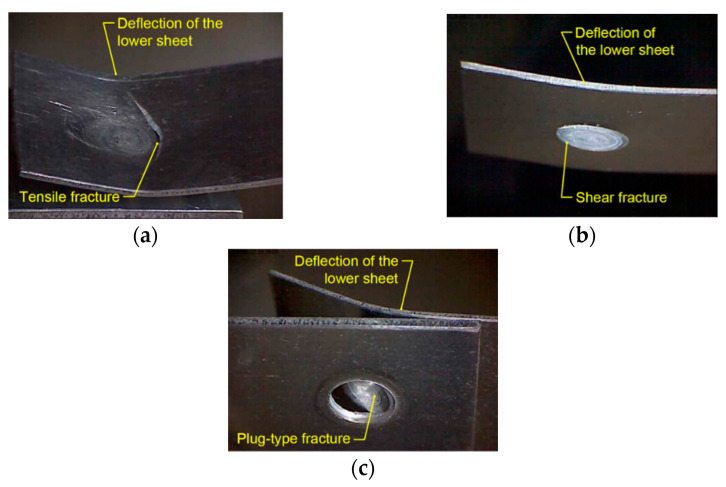
Modes of destruction of a joint subjected to a tensile shear test: (**a**) tensile fracture, (**b**) shear fracture and (**c**) plug-type fracture.

**Figure 11 materials-14-03397-f011:**
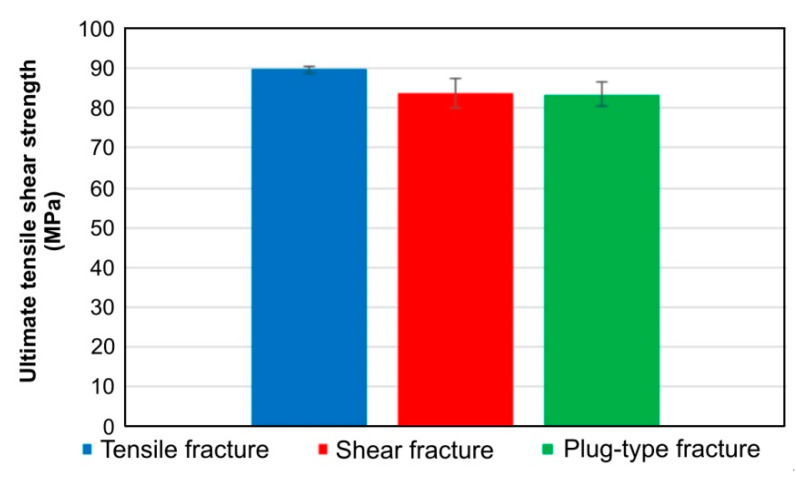
Ultimate tensile shear strength corresponding to specific fracture modes.

**Figure 12 materials-14-03397-f012:**
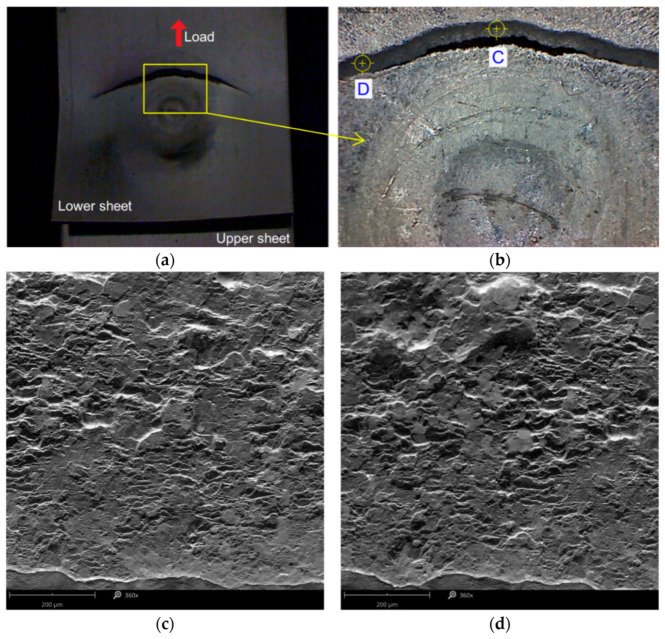
Fracture of the lower sheet which has failed in tensile fracture mode (experiment E2): (**a**,**b**) macroscopic view of the fracture, (**c**,**d**) SEM views of the edge of the fracture in areas C and D indicated in (**b**).

**Figure 13 materials-14-03397-f013:**
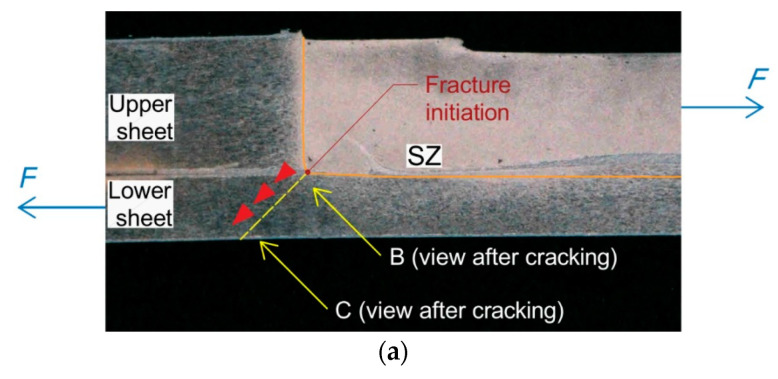
(**a**) Character of the fracture in the lower sheet and (**b**,**c**) SEM micrographs of the fracture surface in the vicinity of the opposite surfaces of the lower sheet.

**Figure 14 materials-14-03397-f014:**
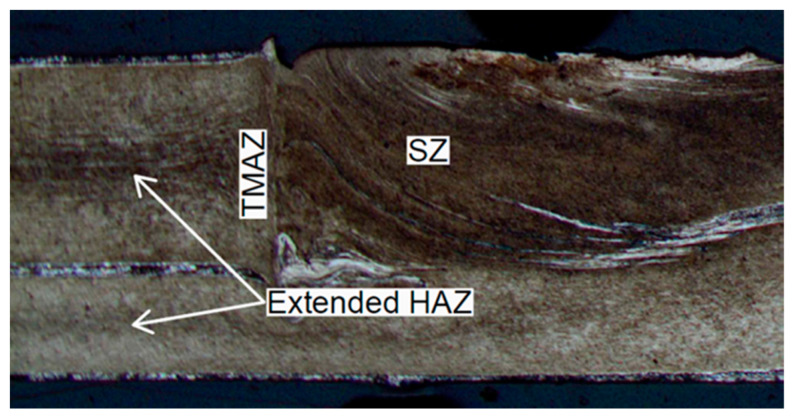
Macrostructure of the weld with extender HAZ (experiment E1).

**Figure 15 materials-14-03397-f015:**
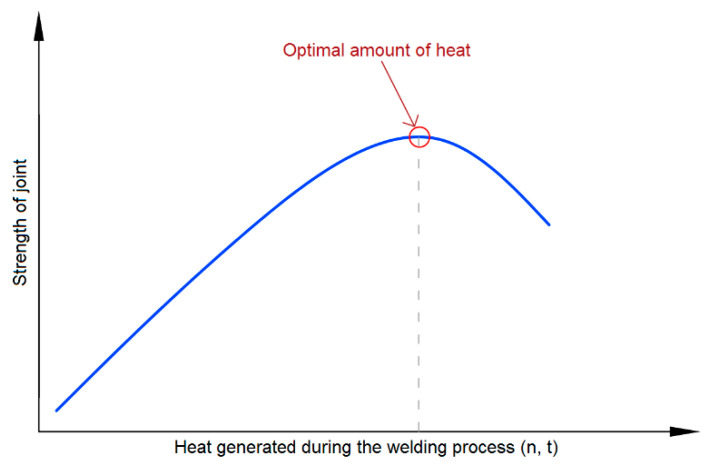
Illustrative diagram of the influence of the amount of heat generated during RFSSW on the strength of the joint.

**Figure 16 materials-14-03397-f016:**
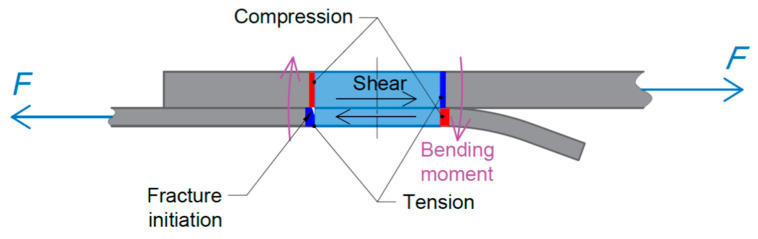
Stresses in a lap joint due to uniaxial tensile force.

**Figure 17 materials-14-03397-f017:**
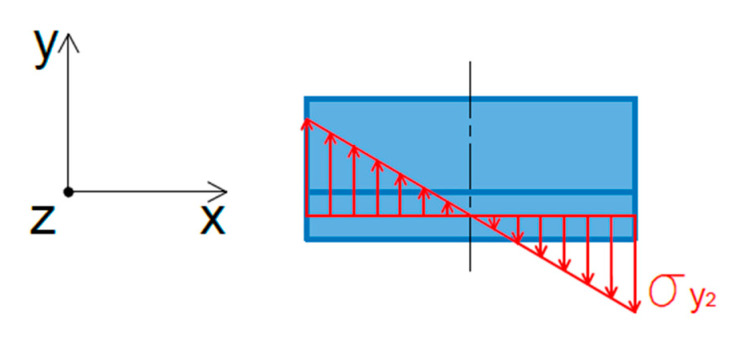
Stresses acting on the weld nugget.

**Figure 18 materials-14-03397-f018:**
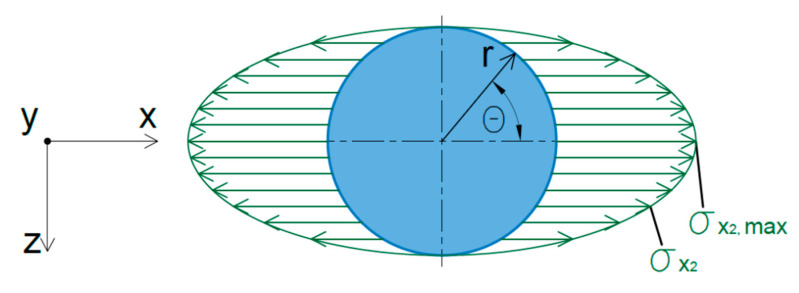
Stress distribution around the weld nugget.

**Figure 19 materials-14-03397-f019:**
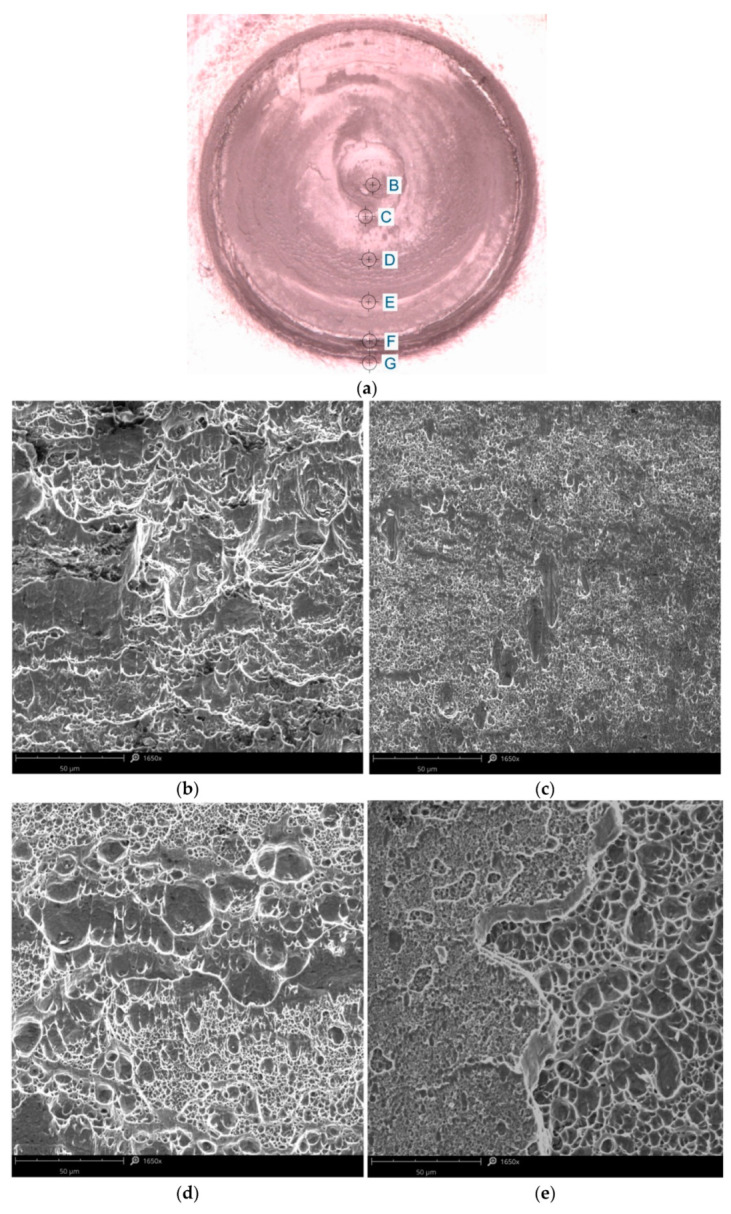
(**a**) fracture surface with the points where SEM micrographs (experiment E3) were prepared indicated, (**b**–**g**) SEM micrographs from the points B–G, respectively.

**Figure 20 materials-14-03397-f020:**
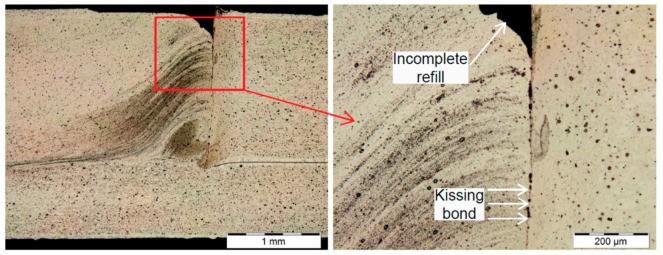
Defects in a weld fabricated with the following conditions: tool rotational speed n = 1800 rpm and duration of welding t = 1.5 s (experiment E4).

**Figure 21 materials-14-03397-f021:**
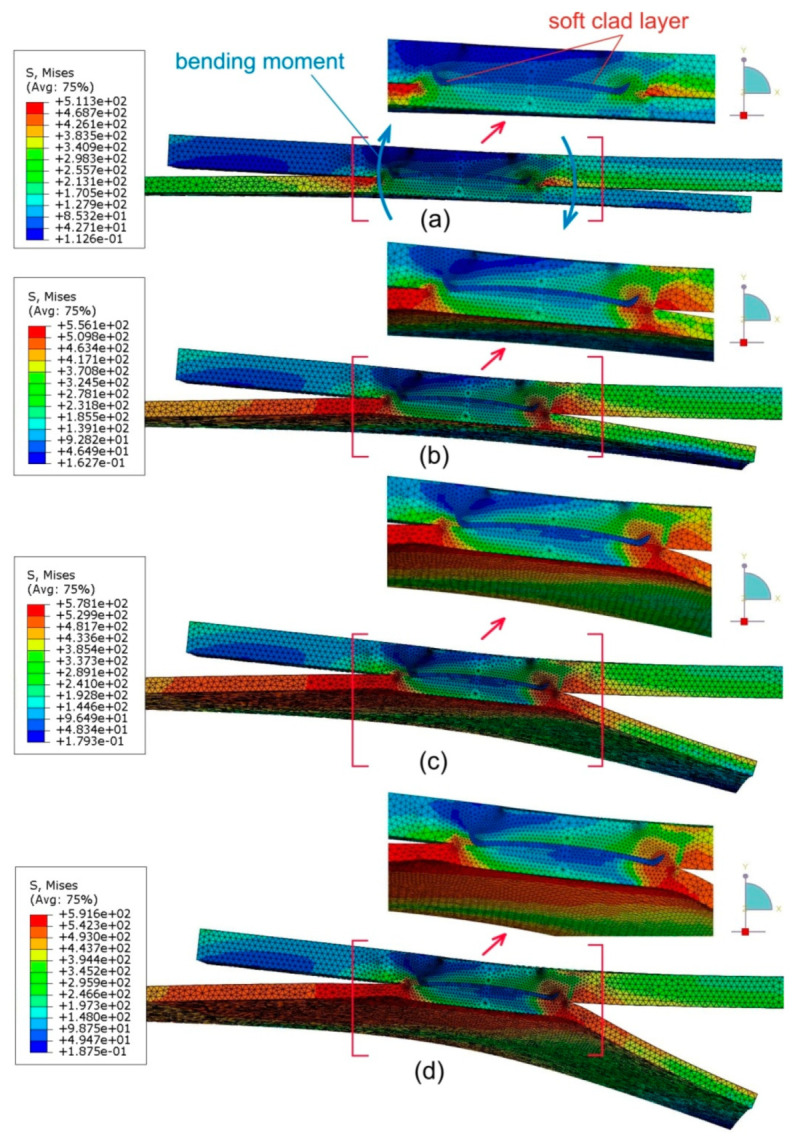
Distribution of equivalent von Mises stress (in MPa) on the longitudinal cross-section of an RFSSW joint under tensile shear loading at different displacements: (**a**) 0.45 mm, (**b**) 0.9 mm, (**c**) 1.35 mm and (**d**) 1.8 mm.

**Figure 22 materials-14-03397-f022:**
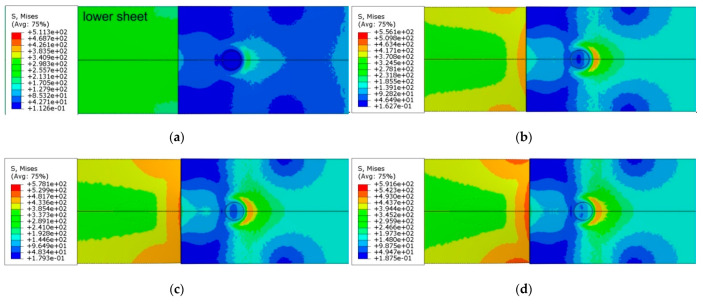
Distribution of equivalent von Mises stress (in MPa) in the RFSSW joint under tensile shear loading at different displacements (view on surface of weld): (**a**) 0.45 mm, (**b**) 0.9 mm, (**c**) 1.35 mm and (**d**) 1.8 mm.

**Figure 23 materials-14-03397-f023:**
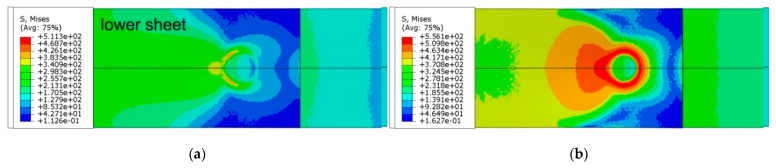
Distribution of equivalent von Mises stress (in MPa) in the RFSSW joint under tensile shear loading at different displacements (view on the bottom of weld): (**a**) 0.45 mm, (**b**) 0.9 mm, (**c**) 1.35 mm and (**d**) 1.8 mm.

**Figure 24 materials-14-03397-f024:**
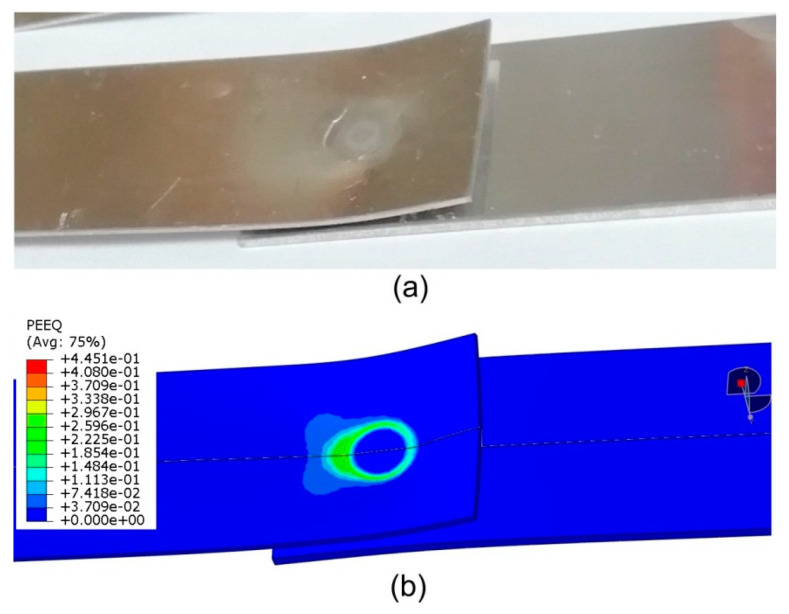
Comparison of the view of specimen (**a**) fabricated at tool rotational speed n = 2400 rpm and duration of welding t = 3.5 s (experiment E2) with (**b**) result of the distribution of plastic strain magnitude; displacement 1.8 mm.

**Figure 25 materials-14-03397-f025:**
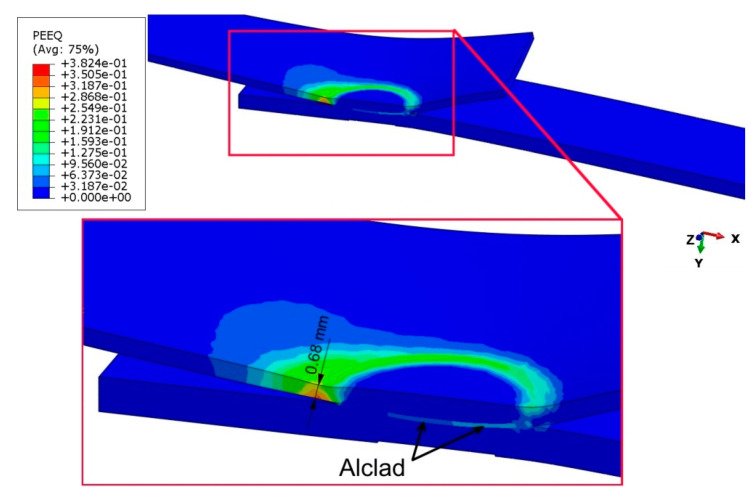
Distribution of equivalent plastic strain on the longitudinal cross-section of an RFSSW joint (experiment E2) under tensile shear loading at a displacement of 1.9 mm.

**Figure 26 materials-14-03397-f026:**
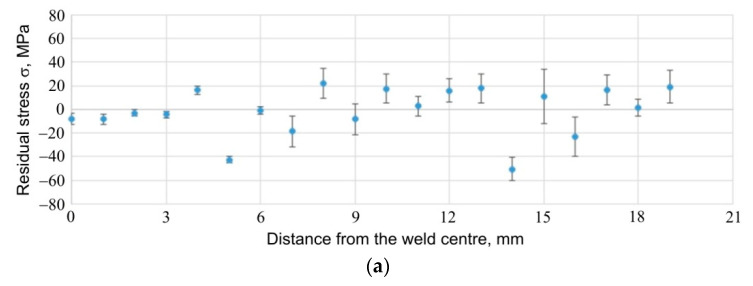
Stresses present in and around the RFSSW joint for specimen E5, for 1.6 mm thick sheet; (**a**) tangential and (**b**) radial component of stresses.

**Figure 27 materials-14-03397-f027:**
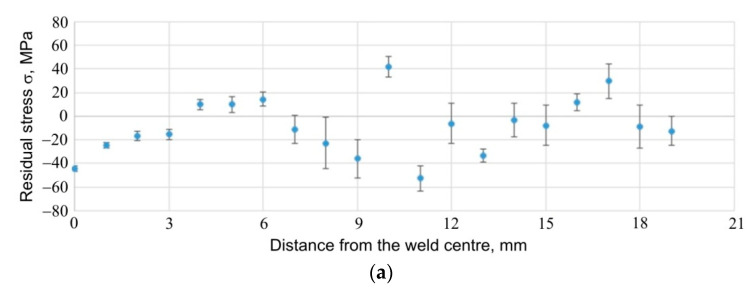
Stresses present in and around the RFSSW joint for specimen E5, for 0.8 mm thick sheet; (**a**) tangential and (**b**) radial component of stresses.

**Figure 28 materials-14-03397-f028:**
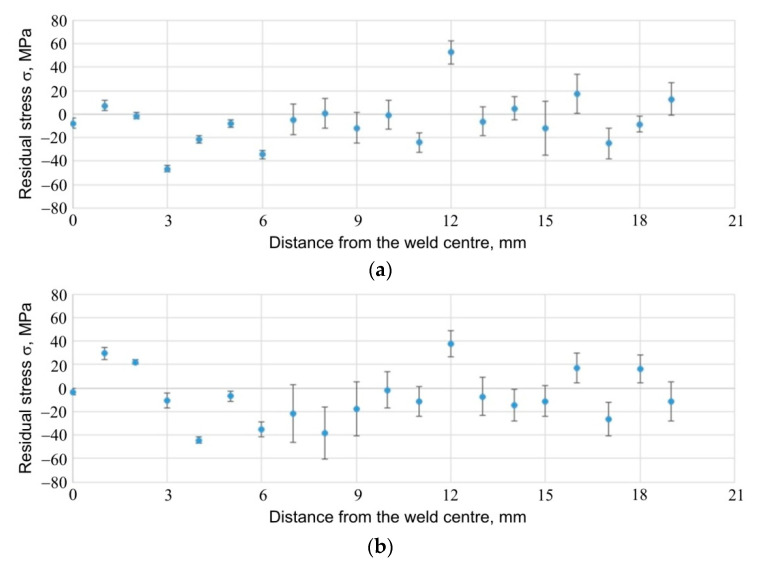
Stresses present in and around the RFSSW joint for specimen E6, for 1.6 mm thick sheet; (**a**) tangential and (**b**) radial component of stresses.

**Figure 29 materials-14-03397-f029:**
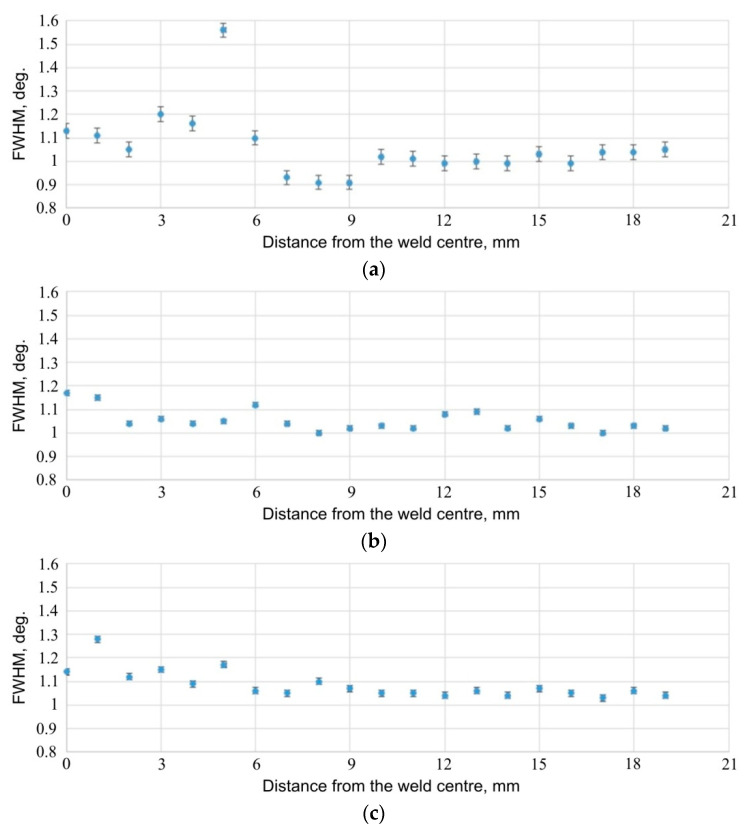
FWHM of diffraction peaks obtained from (**a**) 1.6 mm sheet of E5 specimen, (**b**) 0.8 mm sheet of E5 specimen and (**c**) 1.6 mm sheet of E6 specimen.

**Table 1 materials-14-03397-t001:** Basic mechanical properties of 7075-T6 aluminium alloy *.

*R_p_*_0,2_ (MPa)	*R_m_* (MPa)	*A*_50_ (%)	*K* (MPa)	*n*	*E*, GPa
413.7	482.6	7	610.9	0.045	70.836

* *R_p_*_0,2_—yield stress, *R_m_*—ultimate tensile stress, *A*_50_—elongation.

**Table 2 materials-14-03397-t002:** Experimental plan.

Number of Experiment	Tool Rotational Speed *n* (rpm)	Duration of Welding t (min)
E1	2800	3.5
E2	2400	3.5
E3	2000	2.5
E4	1800	1.5
E5	2200	2.5
E6	2800	1.5

**Table 3 materials-14-03397-t003:** Young’s modulus and yield stress in the principal zones of RFSSW joint.

Zone	Young’s Modulus E, GPa	HV0.1	Yield Stress *σ_y_*, MPa
SZ	89.937	212.3	452.4
TMAZ	89.235	206.0	446.6
HAZ	89.974	195.1	436.7

## Data Availability

The data presented in this study are available upon request from the corresponding author.

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
