# Peer review of "Investigation into the Effect of RFSSW Parameters on Tensile Shear Fracture Load of 7075-T6 Alclad Aluminium Alloy Joints"

_materials, 2021, doi:10.3390/ma14123397_

Round 1

Reviewer 1 Report

The manuscript contains an interesting study about alloys that are generally difficult to weld, but there are several editorial and technical changes that need to be made before it is ready for publication.

1) Be sure to define Rp0,2, Rm, and A in text before you use them in your Table 1.

2) The process you describe does not mention what happens with the soft/molten metal in your description. Please revise it so that this part is explained.

3) Why was the welding time not kept constant for all rotation speeds in your experimental plan?

4) Why only choose one sample to conduct microhardness tests?

5) It seems the specimens chosen for the hardness and XRD tests were chosen without specific justification. Please explain in your methods section why these particular parameters were chosen for testing.

6) What is I1 and I3 in section 2.7?

7) Throughout your Results and Discussion section, you don’t use the experiment labels (e.g. E1, E2, etc.) even though you talk about them in your Materials and Methods section. It makes your discussion harder to follow. Please use the labels you established in your Materials and Methods section, including in your figures.

8) I recommend referencing Sokoluk, M., Cao, C., Pan, S. et al. Nat Commun 10, 98 (2019) since it deals with welding difficult to weld alloys such as Al 7075 in your introduction.

9) This is a minor point, but you should report the UTS for the joints you made in this study not just the maximum load.

10) Please spend a little bit more time discussing the difference between RFSSW and FSSW. This is an important point that explains how your study is unique.

Reviewer 2 Report

Comments for authors are in the attachment.

Reviewer 3 Report

The manuscript presented a novel study on effect on different process parameters on RFSSW joints. Several different process parameters were examined, and it was concluded that the process parameters have direct link with the joint strength and residual stress profile. A variety of literature was surveyed to establish the need for this research; however, the experimental plan is not well described. As a result, it was hard to follow the manuscript.

The abstract needs to be rewritten, mentioning the key outcome of this research.

Page 1 line 41: The statement is a bit confusing. For light weight joining self-piercing riveting (SPR) is getting popularity which also does not require pre hole. This statement needs modification. Additionally, it is also a good idea to include in the introductory discussion about other joining techniques such as clinching and SPR in order to attract more readers who are working on metal joining for light weighting. The following two articles may be cited as well:

Research on the Influence of the AW 5754 Aluminum Alloy State Condition and Sheet Arrangements with AW 6082 Aluminum Alloy on the Forming Process and Strength of the ClinchRivet Joints, Materials, 2021.

Quality of self-piercing riveting (SPR) joints from cross-sectional perspective: A review- Archives of Civil and Mechanical Engineering, 2018.

Page 21 line 483: To be consistent throughout the manuscript, please replace figs with figures.

In the experimental plan 4 different joining condition were mentioned namely from E1 to E4. However, later on, suddenly, in the residual stress section two different joining condition namely I1 and I3 were investigated. Why different conditions were chosen for residual stress measurement in compare with the tensile strength? To make the paper more interesting and readable please include all the joining condition in the experimental section and please be consistent to name them. Please choose either E1… or I1… not both.

What will happen if you change the sheet material order. For example, if you place the 0.8mm sheet on the top do you expect to see a completely different residual stress profile? The research is incomplete without that experiment.

Please rename the x-axis for figures 25-28 as distance from the weld centre. A very similar results were found for the following article: Residual stress profiles in riveted joints of steel sheets, Science and technology of welding and joining, 2015. Although those results are for SPR joints, but it is good to cite that to increase the validity of the current research.

Please show the radial and tangential direction somewhere in the experimental section.

Page 7 line 227: What was the depth set for the x-ray measurement?

Figure 15 is unnecessary as it is already represented in figure 17.

Figures 20-22: Are the stresses in MPa? Please include the unit in the y-axis.

Round 2

Reviewer 1 Report

This manuscript is ready for publication after two minor revisions:

  1. Your explanation for question “Why was the welding time not kept constant for all rotation speeds in your experimental plan?” in the original review is good. Please add it or a condensed version of your response to your Materials and Methods section.
  2. You are missing a “kN” after 5.67 on page 16 of your manuscript.

Author Response

Thank you very much for your valuable comments. The paper has been completed. Information added in the manuscript are highlighted in green.

Reviewer 2 Report

Accept.

Author Response

Thank you very much for your valuable comments.

Reviewer 3 Report

The authors made efforts to improve the article by addressing all the issues raised by the reviewers. It can be publish now.

Author Response

(The authors gave the same response as above.)
